# Performance of a Chest Radiography AI Algorithm for Detection of Missed or Mislabeled Findings: A Multicenter Study

**DOI:** 10.3390/diagnostics12092086

**Published:** 2022-08-28

**Authors:** Parisa Kaviani, Subba R. Digumarthy, Bernardo C. Bizzo, Bhargava Reddy, Manoj Tadepalli, Preetham Putha, Ammar Jagirdar, Shadi Ebrahimian, Mannudeep K. Kalra, Keith J. Dreyer

**Affiliations:** 1Department of Radiology, Massachusetts General Hospital, Harvard Medical School, Boston, MA 02114, USA; 2MGH & BWH Center for Clinical Data Science, Boston, MA 02114, USA; 3Qure.ai, Mumbai 400063, India; 4Internal Medicine, Icahn School of Medicine at Mount Sinai, Elmhurst Hospital Center, Elmhurst, NY 11373, USA

**Keywords:** chest X-ray, artificial intelligence, missed findings, mislabeled, radiology

## Abstract

**Purpose**: We assessed whether a CXR AI algorithm was able to detect missed or mislabeled chest radiograph (CXR) findings in radiology reports. **Methods**: We queried a multi-institutional radiology reports search database of 13 million reports to identify all CXR reports with addendums from 1999–2021. Of the 3469 CXR reports with an addendum, a thoracic radiologist excluded reports where addenda were created for typographic errors, wrong report template, missing sections, or uninterpreted signoffs. The remaining reports contained addenda (279 patients) with errors related to side-discrepancies or missed findings such as pulmonary nodules, consolidation, pleural effusions, pneumothorax, and rib fractures. All CXRs were processed with an AI algorithm. Descriptive statistics were performed to determine the sensitivity, specificity, and accuracy of the AI in detecting missed or mislabeled findings. **Results**: The AI had high sensitivity (96%), specificity (100%), and accuracy (96%) for detecting all missed and mislabeled CXR findings. The corresponding finding-specific statistics for the AI were nodules (96%, 100%, 96%), pneumothorax (84%, 100%, 85%), pleural effusion (100%, 17%, 67%), consolidation (98%, 100%, 98%), and rib fractures (87%, 100%, 94%). **Conclusions**: The CXR AI could accurately detect mislabeled and missed findings. **Clinical Relevance**: The CXR AI can reduce the frequency of errors in detection and side-labeling of radiographic findings.

## 1. Introduction

Chest radiography (CXR) is the most common imaging test, representing up to 20% of all types of imaging procedures [1]. Studies have reported that 236 CXRs are performed per 1000 patients per year, representing up to 25% of annual diagnostic imaging procedures [2]. In 2010 alone, of the 183 million radiographic procedures in the United States on 15,900 radiologic units, CXRs represented almost half of all radiographic images (44%) [3]. Easy accessibility, portability, familiarity, and affordability (relative to other imaging tests) are all factors leading to its widespread use in medical practice for various cardiothoracic ailments [4,5]. Despite their common use, CXRs are difficult to read, and are subject to substantial inter- and intra-reader variations.

To the best of our knowledge, there are no prior publications on the impact of an AI algorithm on addended mislabeled or misinterpreted CXR reports in routine clinical practice. Therefore, we investigated whether a CXR AI algorithm is able to detect missed or mislabeled CXR findings in radiology reports. Section 2, Section 3, Section 4 and Section 5 present, in order, the materials and methods, results, discussions, and conclusions of our study.

### Related Work

A prior retrospective study documented that inter-radiologist and physician concordance were 78% for CXRs [6]. Other studies have reported disagreements between radiologists on CXR findings [7]. CXRs have a high misinterpretation rate, reportedly as high as 30% [8]. The impact of missed radiographic findings is non-trivial. A 1999 study reported that 19% of lung cancers which presented as pulmonary nodules on CXRs were missed [9]. Such missed findings can be catastrophic for both patients and reporting physicians. The Institute of Medicine (IOM) states that 44,000–98,000 patients die in the United States every year because of preventable errors [10].

With the increasing use and availability of approved (for instance, by the US Food and Drug Administration) artificial intelligence (AI) algorithms for several CXR findings [11] (Table 1), we hypothesize that AI could help reduce the frequency of mislabeled or misinterpreted CXRs. Apart from improved interpretation efficiency, several studies have reported improved accuracy of interpretation of several CXR findings [12,13,14]. There are an increasing number of AI algorithms for triaging and detecting different CXR findings, including pneumonia, pneumothorax, pleural effusion, and pulmonary nodules [12,13].

## 2. Materials and Methods

### 2.1. Approval and Disclosures

The institutional review board at Massachusetts General Brigham approved our retrospective study (IRB protocol number: 2020P003950) with a waiver of informed consent. A study co-investigator (SRD) received a research grant from Qure.AI but did not participate in data collection, study evaluation or statistical analysis. Another study co-investigator (MKK) received research grants for unrelated projects from Siemens Healthineers (Erlangen, Germany), Riverain Tech (Miamisburg, OH, USA), and Coreline Inc. (Santa Clara, CA, USA). The remaining co-authors have no financial disclosures. All study authors had equal and unrestricted access to the study data.

### 2.2. Chest Radiographs

We queried multi-institutional radiology search databases totaling 13 million reports to identify all CXRs reports with addenda from 1999–2021. We used the following keywords for search criteria: chest radiograph and addendum. The identified CXRs reports included both portable and posteroanterior upright CXRs. The two search engines used in our study were mPower (Nuance Inc., Burlington, MA, USA) and Render (proprietary institutional search engine). Duplicate CXR reports were excluded when they had the same medical record and accession numbers. The search identified a total of 3469 unique CXR reports between January 2015 to March 2021, as the patients’ medical information was not recorded electronically in our database before 2015. The inclusion criteria were: availability of DICOM CXR images from seven hospitals in our healthcare enterprise (Massachusetts General Hospital (MGH), Brigham Women Hospital (BWH), Faulkner Health Center (FH), Martha’s Vineyard Hospital (MVH), Salem Hospital (NSMC), Newton-Wellesley Hospital (NWH), and Spaulding Rehabilitation Hospital (SRH)) and an addendum with either a missed or side-mislabeled finding. To protect institutional identity (as a higher number of addenda was linked to a greater volume of reported CXRs and not to the quality of reporting), we blinded the names of individual sites before analysis. Missed findings in CXR reports were defined as those reports where there was a missed radiographic finding in the original CXR report which was subsequently corrected with an addendum. Mislabeled findings included wrong side labels of CXR findings in the initial CXR reports which were corrected with addenda. 

Two study coinvestigators (MKK, with fourteen years of experience as a thoracic radiologist and PK, with two years of post-doctoral research in radiology) excluded CXR reports with addenda documenting typographical errors (*n* = 782), wrong report templates (*n* = 341), missing section signoffs (*n* = 289), and communication errors (*n* = 1174). Duplicate reports of the same exam and patient were excluded as well (*n* = 302). Other exclusion criteria were addenda concerning findings that could not be assessed with the AI algorithm, such as cardiac calcification (*n* = 27), diaphragmatic hernia (*n* = 81), clavicle (*n* = 29) or humeral (*n* = 34) fractures or dislocations, lines (*n* = 94), and devices (*n* = 37). The final sample size after the application of inclusion and exclusion criteria was 279 CXRs with addenda belonging to 279 patients; the patient demographics are summarized in the results section. Figure 1 summarizes the distribution of missed and mislabeled CXR findings included in our study.

The study co-investigators noted the organization name for addended CXR reports and the specific missed finding name, such as pneumothorax, nodule, atelectasis, rib fractures, and pleural effusion. A chest radiologist (MKK) reviewed all CXRs with missed findings/labeled nodules and assessed their size and clinical importance on a three-point scale (1: not significant, the nodule is definitely benign such as granuloma; 2: indeterminate clinical importance; 3: definitely of clinical importance). Information on request for further imaging or follow-up evaluation was recorded from the reports. 

### 2.3. AI Algorithm

All 279 frontal CXRs were deidentified, exported as DICOM images, and processed with an offline AI algorithm (qXR, Qure.AI, Mumbai, India) installed on a personal computer within our institutional firewall to protect patient privacy. Although approved in several countries in Asia, Africa, and Europe, the AI algorithm used in our study is not cleared by the US FDA. The algorithm was trained on over 3.7 million CXRs and radiology reports from various healthcare sites from different parts of the world. The algorithm uses a series of convolutional neural networks (CNNs) trained to identify different abnormalities on frontal CXRs. The algorithm first resizes and normalizes CXRs to decrease variations in the acquisition process, then applies modifications in either densenet or resnet network architectures to separate CXRs from radiographs of other anatomies. Subsequently, multiple networks, including densenets and resnets, are applied for individual CXR findings. Further technical details of the algorithm have been described in prior publications [15]. The algorithm was validated on a separate dataset of over 93,000 CXRs from multiple sites in India. Neither training nor validation datasets included CXRs from any test sites. The image algorithms (qXR v3) were trained and tuned on a training set of 3.7 million chest X-rays with the corresponding reports. Optimal thresholds were selected using a proprietary method developed at Qure.ai, along with standard methods such as Youden’s Index. These thresholds were additionally validated on a test set of over 93,000 CXRs which was not used during training. The algorithm is an ensemble of more than fifty models, each used for detection of specific abnormalities or features in CXRs. Multiple architectures are selected for use during training, each with a different number of layers/parameters. The algorithms use different architectures, including Efficientnet-b5/6/7, Resnet 50/101d, and ResneXt101. In general, the models have parameter counts ranging from 20–50 million. The common optimizers include SGD and ADAM, and models are trained for around 200 epochs. The learning rate schedulers vary from model to model. The process used has been previously described in work done at Qure.ai with qXR, which can be found at: https://arxiv.org/abs/1807.07455 (accessed on 19 July 2018). The threshold values used were part of the commercial version of qXR and were frozen before the start of the study. The algorithm takes less than 10 s to process each CXR.

### 2.4. Statistical Analyses

Statistical analysis was performed with Microsoft EXCEL (Microsoft Inc., Redmond, WA, USA). To assess the performance of the AI algorithm, we predefined true positive (i.e., the specific missed finding is identical in addendum and AI output; for example, both the addendum and AI output document pneumothorax), true negative (addendum and AI output agree on the absence of specific findings; for example, both addendum and AI agree on the absence of pneumothorax), false positive (addendum or the original radiology report did not document a finding identified by the AI algorithm; for example, the AI identified pneumothorax which was not present in the radiology report or the CXR), and false negative (the addendum described a missed finding did not correspond to that detected by the AI; for example, the addendum documented the presence of pneumothorax which the AI did not detect) The sensitivity, specificity, accuracy, and receiver operating characteristics (ROC) with the area under the curve (AUC) were calculated using Microsoft Excel 16 (Microsoft Inc., Redmond, WA, USA) and SPSS version 26 (IBM Inc., Chicago, IL, USA).

## 3. Results

Of the 279 CXRs with addenda performed in 279 patients (mean age 59 ± 20 years), 143 belonged to male patients and 136 to female patients. There were 230 PA CXRs and 49 portable CXRs in the dataset. As the algorithm labeled both pneumonia and atelectasis as consolidation, we reported the sum of these two findings as consolidation. Table 2 summarizes the distribution of missed and mislabeled CXR findings in our study according to CXR types (posteroanterior or portable). Regardless of the sites, most missed and mislabeled findings in the addenda were present in reports of posteroanterior CXRs as compared to portable CXRs (*p* < 0.001). Documentation of missed findings greatly outnumbered mislabeled findings. The most common missed findings included pneumothoraces (100/279; 35.8%), consolidation (62/279; 22.2%), pulmonary nodules (54/279; 19.4%), rib fractures (48/279; 17.2%), and pleural effusions (15/279; 5.4%).

The sensitivity, specificity, accuracy, and AUCs of the AI algorithm for different findings are summarized in Table 3. The AI’s highest performance was in the detection of pulmonary nodules and consolidation, and was the lowest in pleural effusion (low specificity). Table 4 illustrates the frequency of true positive, true negative, false positive, and false negative results for each finding.

Figure 2 summarizes the AUCs of the algorithm for different radiographic findings. There was no significant difference in the performance of the AI on CXRs from sites A and B or between the portable and posteroanterior CXRs (*p* > 0.5). Data from the remaining sites were not compared due to low sample sizes (<5 data points per finding) for missed and mislabeled findings. The AI algorithm had a moderate to high AUC for all missed findings. For true positives, the AI annotated heat map showed the lesion as being on the correct side, instead of the mislabeled findings reported in the addended radiology reports. Figure 3 displays different missed findings which were correctly detected (true positive) by the AI algorithm.

Among the 279 CXR, 41 CXR addenda requested for additional assessment: rib fractures (*n* = 4), pneumothoraces (*n* = 3), and pulmonary nodules (*n* = 34). All patients with missed pneumothoraces had chest tube placement. All 34 CXRs with missed nodules underwent chest CT, three patients underwent lung nodule biopsy (one benign nodule, two malignant nodules). The 31 remaining pulmonary nodules remained stable on follow-up imaging (CXR and/or CT). Figure 4 illustrates the distribution of missed nodule size and further evaluation.

## 4. Discussion

Here, we report the frequency of different CXR findings which were either missed or mislabeled and later corrected with an addendum. The assessed AI algorithm can help to identify such findings and errors with high performance (AUC 0.82–0.99). Although prior studies have described the comparable performance of AIs either standalone or a second reader [21], most studies have evaluated consecutive or selected CXRs without specific attention to the clinical significance of detected or missed radiographic findings.

Prior research publications have investigated the frequency of misdiagnosis of CXR findings [22]. In a recent study, Wu et al. described an AI algorithm that reaches and exceeds the performance of third-year radiology residents for detecting findings on frontal chest radiographs, with a mean AUC of 0.772 for the assessed AI algorithm [14]. Such AI algorithms can improve accuracy while improving the workflow efficiency of reporting. In our institution, addenda are issued for final signed-off radiology reports by the attending radiologists or a fully licensed trainee such as a clinical fellow. Therefore, high accuracy (up to 0.99) of the assessed AI algorithm pertains to findings missed by interpreting physicians beyond residency training.

The high accuracy and AUCs of the AI algorithm for detecting consolidation and pulmonary nodules in our study correspond to those reported in recent studies. Behzadi-Khormouji et al. reported an accuracy of 94.67% for detecting consolidation on CXRs using an AI model [21]. Likewise, the performance of our AI algorithm for detecting all-cause pulmonary nodules is comparable to the overall performance of another AI algorithm (Lunit Inc., Seoul, Korea). Yoo et al. reported a sensitivity of 86.2% and specificity of 85% for all-cause nodules, and a higher sensitivity of AI (up to 100%) compared to 94.1% for radiologists in detecting malignant nodules on digital CXRs [23]. Similarly, the high sensitivity, specificity, accuracy, and AUC of the AI algorithm for detection of pneumothorax compares well with other multicenter studies using other AI algorithms, such as Thian et al., who reported an AUC of 0.91–0.97 for detection of pneumothorax detection with their AI algorithm [24].

The main implication of our study pertains to the performance and potential use of AI when reporting CXRs. In light of the high volume of CXR use in hospital settings, relatively low reimbursement for CXR interpretation, and pressure for rapid and efficient reporting, compounded by the highly subjective nature of projectional radiography, reporting errors on CXRs are common. In such circumstances, as noted from our study and supported by other investigations [12,13,14], AI algorithms can reduce the frequency of commonly missed and mislabeled CXR findings. Furthermore, routine use of CXR AI in interpretation has the potential to avoid common reporting errors, and therefore reduce the need to issue addenda to previously reported exams. At the same time, AI algorithms can potentially shift the focus from under- or non-reporting of radiographic findings to over-reporting of findings due to high false positive outputs. Such challenges can be addressed with robustly trained AI models and selection of appropriate cut-off values that maintain a good balance of sensitivity and specificity across different radiography units and radiographic quality. Although we did not compare the AI’s performance with other models from the literature, the AUCs of our AI algorithm were similar to those reported for other models assessed using open access CXR datasets (https://nihcc.app.box.com/v/ChestXray-NIHCC/file/220660789610 (accessed on 4 August 2022)) [25]. Apart from detection of radiographic findings assessed in our study with an AI algorithm, other studies have assessed applications of AI for prioritizing interpretation of CXRs in order to expedite reporting of abnormal CXRs and specific findings [26]. Baltruschat et al. reported the use of AI-based worklist prioritization for a substantial reduction in reporting turnaround time for critical CXR findings [27]. Similar improvements in reporting time with worklist prioritization using AI have been reported for other body regions as well, such as head CTs for intracranial hemorrhage [28,29]. Finally, although our study highlights how AI could help to reduce missed findings and errors in radiology reports, larger studies with greater and more balanced representation of each missed finding are important before generalizing our observations to other sites and practices. This is particularly true for CXRs with pulmonary nodules and pleural effusions, due to their sparse distribution in our study datasets.

There are several limitations to our study. Although we queried over 13 million radiology reports from 1999 to March 2021 to identify 3469 CXR reports with addenda, the stringent inclusion and exclusion criteria made our sample size small (*n* = 279 CXRs), which is the primary limitation of our study. While a larger dataset for AI algorithm is ideal, our study provides a representative snapshot of missed and mislabeled findings on consecutive eligible CXRs from multiple sites. While addended radiology reports describing missed and mislabeled findings pertain to identified or recognized errors in reporting, they underestimate the true incidence of missed or mislabeled findings, as most findings might not be discovered or corrected in subsequent follow-up CXRs or other imaging tests such as CT. The purpose of our study was not to uncover the true incidence of missed CXR findings, rather, it was to investigate the performance of the AI algorithm on missed or mislabeled findings deemed important by the referring physicians and/or radiologists, and thus addressed with addenda. Due to the limited sample size, we were unable to determine the performance of the AI for missed rare findings such as mediastinal and hilar abnormalities, cavities, and pulmonary fibrosis. In addition, the performance of AIs can vary based on the type of findings; therefore, our results may not be generalizable to sites with different distributions of CXR findings.

Another limitation of our study pertains to findings that are currently not detected by the AI algorithm, such as placement of lines and devices, which was a significant contributor of excluded portable CXR reports with addenda. Although CXRs included in our study belonged to real-world CXRs with addended reports to add missed findings, we did not include real-world randomized CXR datasets without addended reports, as prior studies have reported on performance (sensitivity, specificity, AUC, and accuracy for individual CXR findings) of the AI algorithm used in our study on such datasets [24]. We did not specifically perform a systematic analysis of the algorithm in order to understand domain bias at the study sites. However, the inclusion of different study site types (including quaternary, community, cottage, and rehabilitation hospitals), different radiographic equipment, and a large group of interpreting radiologists would have minimized such bias in our study. Finally, we did not process the CXRs with other AI algorithms, and therefore we cannot comment on the relative performance of different AI algorithms.

## 5. Conclusions and Future Work

In conclusion, our study demonstrates that AI can help to identify several missed and mislabeled findings on CXRs. As a secondary reader, the assessed AI algorithm can help radiologists to identify and avoid common mistakes in detection, description, and labeling of specific radiographic findings, including consolidation, pulmonary nodules, pneumothorax, rib fractures, and to a lesser extent, pleural effusions. Future studies with a larger number of reporting errors can help to assess the effectiveness and relative performance of AI algorithms in improving the accuracy of radiology reporting for chest radiographs.

## Figures and Tables

**Figure 1 diagnostics-12-02086-f001:**
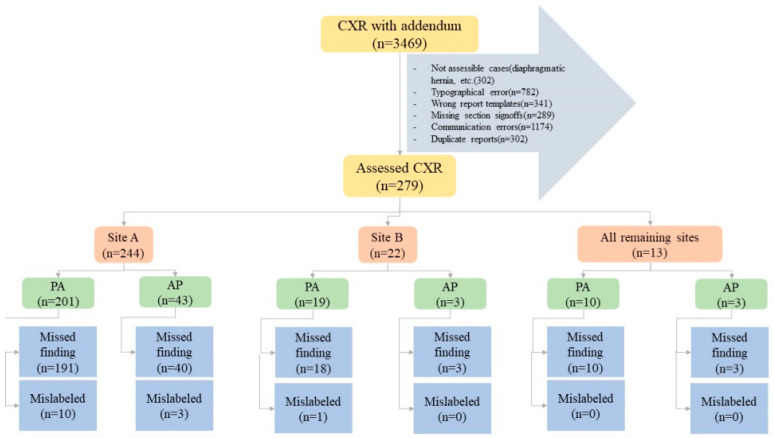
Flow diagram summarizing the study methods and distribution of specific missed and mislabeled findings at different sites and CXR types (PA—posteroanterior CXR; port—portable CXR).

**Figure 2 diagnostics-12-02086-f002:**
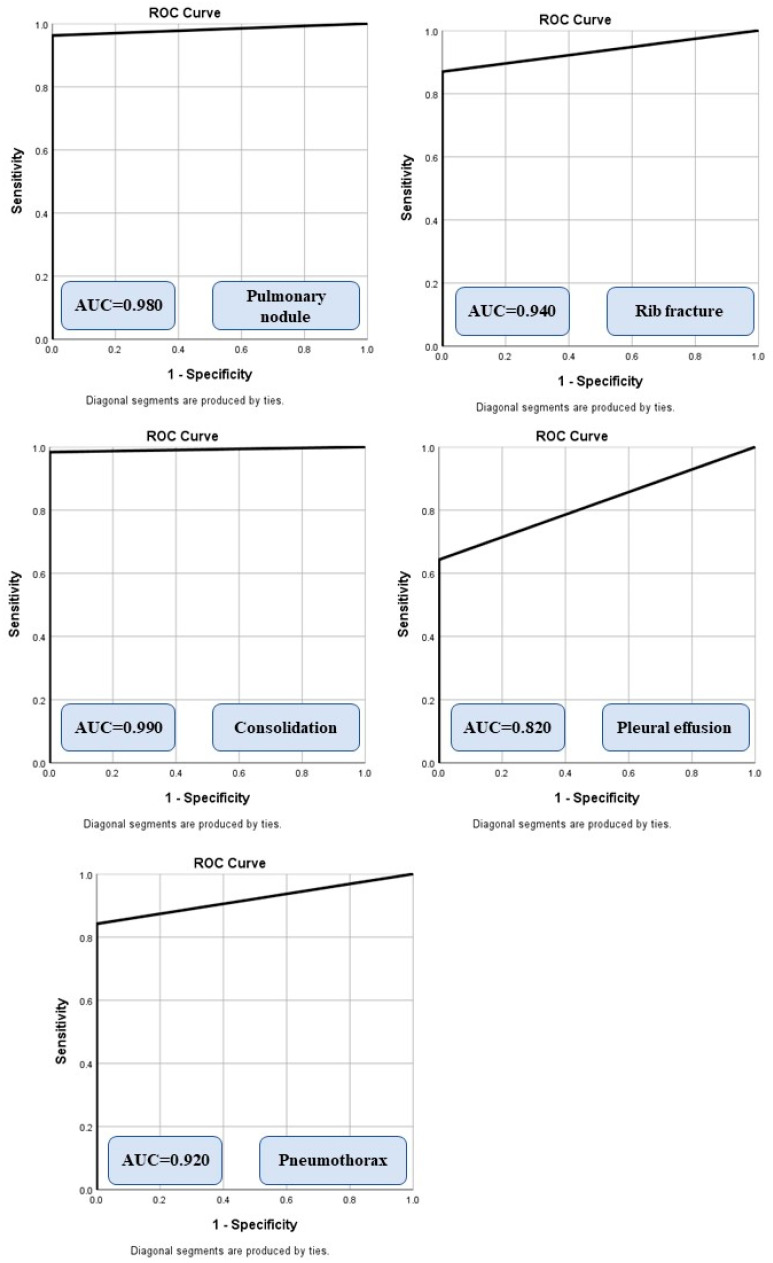
Receiver operating characteristic analyses with area under the curve (AUC) for different missed findings detected by the AI algorithm.

**Figure 3 diagnostics-12-02086-f003:**
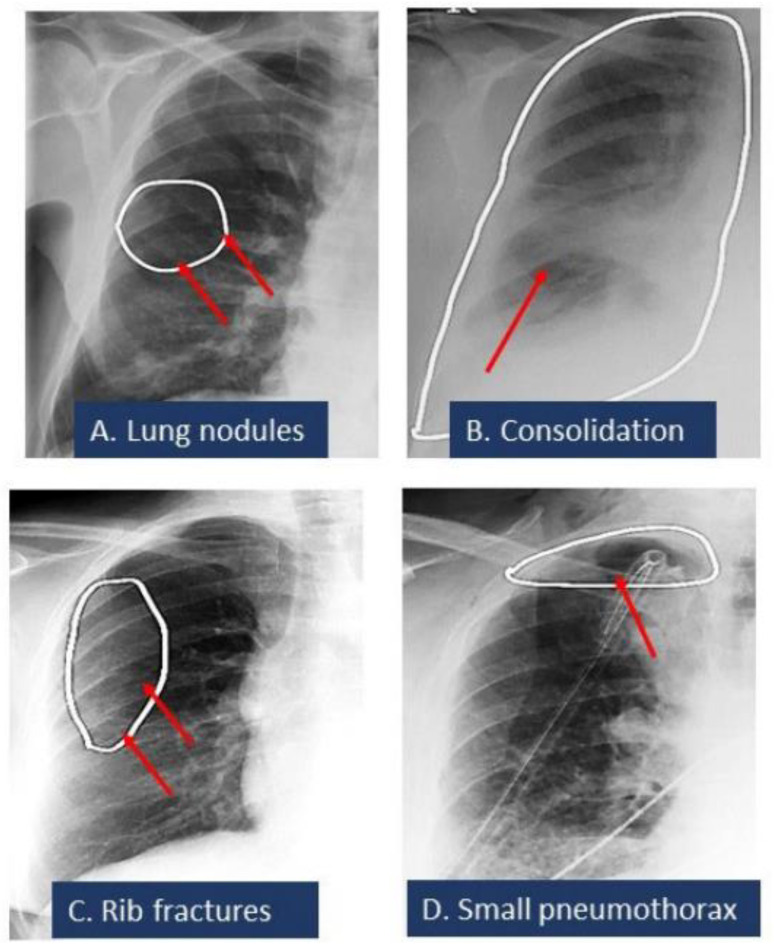
Spectrum of missed CXR findings: pulmonary nodule (**A**), pneumonia (**B**), rib fracture (**C**), and pneumothorax (**D**) for which radiologists issued addenda to their original radiology reports. These findings were detected by the AI algorithm. White ellipses illustrated the location of findings annotated by AI algorithms. Red arrows pointed out the finding.

**Figure 4 diagnostics-12-02086-f004:**
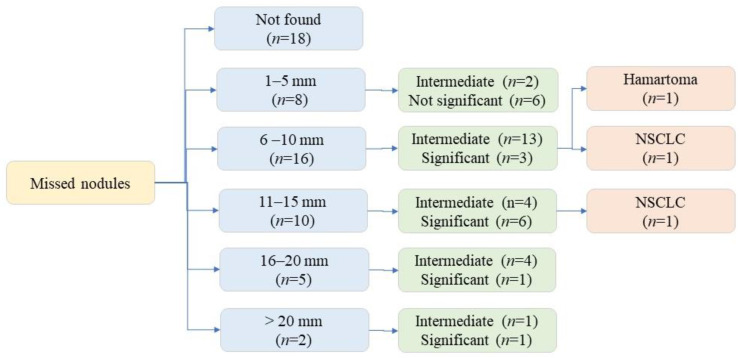
Flow diagram illustrating missed nodule distribution based on size and significance.

**Table 1 diagnostics-12-02086-t001:** Summary of recent studies on use of AI for CXRs and chest CT support, expanding applications, and growing evidence for its use in clinical practice.

Authors (Year)	Sample Size and Approach	Results
Lan et al. (2022) [15]	60 Chest CTs assessed both manually and with AI assistance	Unaided false positive (FP) rate was 0.617–0.650/CT and sensitivity was 59.2–67.0%; with AI assistance, the FP was 0.067–0.2/CT and the sensitivity was 59.2–77.3%
Zhang et al. (2022) [16]	860 chest CT screenings assessed by 14 residents and 15 radiologists; in addition, one radiologist and one resident re-evaluated CTs with AI assistance.	The accuracy and sensitivity of radiologists for solid nodules were 86% and 52%, compared to 99.1% and 98.8% with AI-assistance.
Rudolph et al. (2022) [17]	563 CXRs retrospectively assessed by multiple radiologists and compared with an AI system.	AI-assisted interpretation mimicked the most sensitive unassisted interpretation, with AUCs of 0.837 (pneumothorax), 0.823 (pleural effusion), and 0.747 (lung lesions)
Nguyen et al. (2022) [18]	6285 CXRs for abnormality detection with an AI algorithm	AI had 79.6% accuracy, 68.6% sensitivity, and 83.9% specificity; AI algorithms can help with interpretation of the CXRs as a second reader.
Ajmera et al. (2022) [19]	1012 posteroanterior CXRs for diagnosis of cardiomegaly	An AI algorithm improved sensitivity for identifying cardiomegaly from 40.5% to 88.4%.
Homayounieh et al. (2021) [20]	100 posteroanterior CXRs; detection of pulmonary nodules	Mean detection accuracy of pulmonary nodules increased by 6.4% with AI assistance for different levels of detection difficulty and reader experience.

**Table 2 diagnostics-12-02086-t002:** Summary of missed (MS) and mislabeled (ML) findings in posteroanterior (PA) and portable CXRs at different sites included in our study.

	Site ASite A				Site B				All Remaining Sites			
CXRs	PA		Portable		PA		Portable		PA		Portable	
Findings	MS	ML	MS	ML	MS	ML	MS	ML	MS	ML	MS	ML
Consolidation	49	3	3	0	4	0	0	0	3	0	0	0
Pulmonary nodule	28	1	8	1	8	1	3	0	2	0	2	0
Pneumothorax	68	6	20	1	2	0	0	0	3	0	0	0
Pleural effusion	10	0	0	1	1	0	0	0	2	0	1	0
Rib fracture	36	0	9	0	3	0	0	0	0	0	0	0

**Table 3 diagnostics-12-02086-t003:** Summary statistics of AI performance detecting missed findings on CXRs. The numbers in parentheses represent 95% confidence interval for the area under the curve (AUC).

Findings	Sensitivity	Specificity	Accuracy	AUC
Pulmonary nodule	96	100	96	0.98 (0.94–1.00)
Consolidation	98	100	98	0.99 (0.97–1.00)
Rib fracture	87	100	94	0.94 (0.85–1.00)
Pleural effusion	100	17	67	0.82 (0.54–1.00)
Pneumothorax	84	100	85	0.92 (0.86–0.98)

**Table 4 diagnostics-12-02086-t004:** Summary frequencies of true positive, true negative, false positive, and false negative results of each finding.

Findings	True Positive	True Negative	False Positive	False Negative
Pulmonary nodule	51	1	0	2
Consolidation	62	62	0	1
Rib fracture	20	25	0	3
Pleural effusion	9	1	0	5
Pneumothorax	80	5	0	15

## Data Availability

Not applicable.

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
