# Peer review of "Performance of a Chest Radiography AI Algorithm for Detection of Missed or Mislabeled Findings: A Multicenter Study"

_diagnostics, 2022, doi:10.3390/diagnostics12092086_

Round 1

Reviewer 1 Report

The authors investigated if a CXR AI algorithm can detect missed or mislabeled CXR findings in radiology reports. The sample size is adequate and sufficient for some tasks but not for all tasks in this study. the authors need to elaborate more on this issue.

We also observe imbalanced class distribution in this study and it is not clear how the authors managed this issue before deriving any conclusion.

Author Response

Dear Editor and Reviewers,

Thank you very much for your letter and for the reviewers’ comments concerning our manuscript titled” Performance of a chest radiography AI algorithm for detection of missed or mislabeled findings: A multicenter study” (ID: 1878677). Those comments are all valuable and very helpful for revising and improving our paper, as well as the important guiding significance to our researches. We have studied comments carefully and have made correction which we hope will meet with approval.

Reviewer 1

Comments and Suggestions for Authors

The authors investigated if a CXR AI algorithm can detect missed or mislabeled CXR findings in radiology reports. The sample size is adequate and sufficient for some tasks but not for all tasks in this study. the authors need to elaborate more on this issue.

We also observe imbalanced class distribution in this study, and it is not clear how the authors managed this issue before deriving any conclusion.

Response: We agree with the esteemed reviewer on these two points. We had mentioned this in our limitations section and have further specifically stated this as part of the implication paragraph of the manuscript: “Finally, although our study highlights how AI could help reduce missed findings and errors in radiology reports, larger studies with greater and balanced representation of each missed finding is important before generalizing our observations to other sites and practices. This is particularly true for CXRs with pulmonary nodules and pleural effusions due to their sparse distribution in our study datasets.”

Reviewer 2 Report

1- What about running time (execution time) of the method?

2- The "Related Work" section is missing. A new section should be added between "Introduction" and "Section 2". Some parts of the "Introduction" section can be moved to the new section.  

In addition, providing a table that summarizes the related work would increase the understandability of the difference from the previous studies in the related works section. 

3- The organization of the paper (the structure of the manuscript) may be written at the end of the "Introduction" section. 

For example: "Section 2 presents ... Section 3 gives ...."

4- What are the parameter values of the algorithm? The authors may write parameter setting of CNN.  

For example, number of layers, learning rate, activation function, maximum epoch, optimizer (Adam or another), etc. 

5- The symbols in the text should be italic. 

For example: 

-"typographical errors (n=782) , wrong report templates (n=341), 89 missing section signoffs (n=289), and communication errors (n=1174)"

6- In the reference list, there is no any paper published in 2022.  

I suggest the authors citing the most recent papers (especially published in 2022).

7- Conclusion section is missing.

After "4. Discussion", a new section "5. Conclusion and Future Work" may added. 

Author Response

Dear Editor and Reviewers,

Thank you very much for your letter and for the reviewers’ comments concerning our manuscript titled” Performance of a chest radiography AI algorithm for detection of missed or mislabeled findings: A multicenter study” (ID: 1878677). Those comments are all valuable and very helpful for revising and improving our paper, as well as the important guiding significance to our researches. We have studied comments carefully and have made correction which we hope will meet with approval.

Reviewer 2

Comments and Suggestions for Authors

1- What about running time (execution time) of the method?

Response: We have added the following to the methods section: The algorithm takes less than 10 seconds to process each CXR.

2- The "Related Work" section is missing. A new section should be added between "Introduction" and "Section 2". Some parts of the "Introduction" section can be moved to the new section.  

 Response: We have followed your instructions and added related work section.

In addition, providing a table that summarizes the related work would increase the understandability of the difference from the previous studies in the related works section. 

Response: We have added a table to address this comment from the reviewer. Thank you.

Authors (year)

Sample size and approach

Results

Lan et al. (2022)

600 Chest CT assessed manually and with AI assistance

Unaided false positive (FP) rate was 0.617-0.650/CT, and the sensitivity was 59.2-67.0%.

With AI assistance, the FP was 0.067-0.2/CT, and the sensitivity was 59.2-77.3%

Zhang et al. (2022)

860 chest CT screening assessed by 14 residents and 15 radiologists. In addition, one radiologist and one resident re-evaluated CT with AI-assistance. A reading panel set the gold standard.

Accuracy and sensitivity of radiologists for solid nodules were 86% and 52% as compared to 99.1% and 98.8% with AI-assistance.

Rudolph et al. (2022)

 563 CXRs were retrospectively assessed by multiple radiologists and compared with an AI system.

AI assisted interpretation mimicked the most sensitive unassisted interpretation with AUCs of 0.837 (pneumothorax), 0.823 (pleural effusion), and 0.747 (lung lesions)

Nguyen et al. (2022)

6285 CXRs for detecting abnormality detection with an AI algorithm

AI had 79.6% accuracy, 68.6% sensitivity, and 83.9% specificity. AI algorithm can help in interpretation of the CXRs as a second reader.

Ajmera et al. (2022)

1012 posteroanterior CXRs for diagnosis of cardiomegaly

AI algorithm improved sensitivity for identifying cardiomegaly from 40.5% to 88.4%.

Homayounieh et al (2021)

100 posteroanterior CXRs detection of pulmonary nodules

Mean detection accuracy of pulmonary nodules increased by 6.4% with AI assistance for different levels of detection difficulty and reader experience.

3- The organization of the paper (the structure of the manuscript) may be written at the end of the "Introduction" section. 

For example: "Section 2 presents ... Section 3 gives ...."

 We have added this information as requested.

4- What are the parameter values of the algorithm? The authors may write parameter setting of CNN.  For example, number of layers, learning rate, activation function, maximum epoch, optimizer (Adam or another), etc.  We have added further information on the algorithm as requested. \

We have added following details to the algorithm section. “The algorithm is an ensemble of more than 50 models, each for detection of specific abnormalities or features in CXRs. Multiple architectures are selected for the use during training, each with a different number of layers / parameters. The algorithms use different architectures including Efficientnet-b5/6/7, Resnest 50/101d, and ResneXt101. In general, the models have parameter counts ranging from 20-50 million. The common optimizers include SGD and ADAM and models are trained for around 200 epochs. The learning rate schedulers vary from model to model.”

5- The symbols in the text should be italic. 

For example: 

-"typographical errors (n=782), wrong report templates (n=341), 89 missing section signoffs (n=289), and communication errors (n=1174)"

Response: We made the changes

6- In the reference list, there is not any paper published in 2022.  

I suggest the authors citing the most recent papers (especially published in 2022).

We have updated the reference and included references from 2022.

7- Conclusion section is missing.

After "4. Discussion", a new section "5. Conclusion and Future Work" may be added. 

Response: We made the changes

Round 2

Reviewer 1 Report

All concerns are mentioned in the manuscript. No further comments.